# *Salmonella*-Induced Cell Death in Cancer Immunotherapy: What Lies Beneath?

**DOI:** 10.3390/biomedicines14010012

**Published:** 2025-12-20

**Authors:** Amy Mónaco, Sofía Chilibroste, María Clara Plata, Jose Alejandro Chabalgoity, María Moreno

**Affiliations:** Unidad Académica de Desarrollo Biotecnológico, Instituto de Higiene, Facultad de Medicina, Universidad de la República, Montevideo 11600, Uruguay; amonaco@higiene.edu.uy (A.M.); schilibroste@higiene.edu.uy (S.C.); mplata@higiene.edu.uy (M.C.P.)

**Keywords:** *Salmonella*, cancer immunotherapy, cell death, apoptosis, pyroptosis, autophagy

## Abstract

Bacteria-based cancer immunotherapies are regaining attention due to recent advances in understanding the mechanisms underlying their efficacy, making them promising tools for cancer treatment. Among these, *Salmonella* stands out as one of the most extensively studied microorganisms in this field. Its ability to directly induce tumor cell death while stimulating the immune system offers unique therapeutic advantages, as cell death within an inflammatory environment may enhance the release of tumor antigens and promote effective antitumor immune responses. Although multiple studies have addressed *Salmonella*-induced cell death, the nomenclature and classification of death modalities are often inconsistent—either because earlier reports predate the formalization of certain death pathways, or due to overlapping criteria between different types of cell death. This review aims to comprehensively analyze the available evidence on *Salmonella*-induced apoptosis, pyroptosis and autophagy, as well as other less characterized death modalities. Given that most mechanistics evidence on *Salmonella*-induced cell death has been generated in myeloid cells, we primarily focus on the myeloid compartment while integrating available observations from tumor cells and other immune populations when relevant, organizing the existing data under current definitions and concepts, and highlighting the challenges of manipulating these pathways to optimize bacterial-based immunotherapies.

## 1. Introduction

The use of microorganisms as therapeutic agents in cancer has gained increasing interest over the past two decades. This approach, often referred to as bacterial cancer therapy or microbial-based immunotherapy, exploits the unique ability of certain microbes to preferentially colonize tumor tissues, modulate the tumor microenvironment (TME), and stimulate robust antitumor immune responses. Historically, the concept dates back to the late 19th century with the work of Dr. William Coley, who used heat-killed *Streptococcus pyogenes* and *Serratia marcescens* to treat sarcomas—an early form of immunotherapy now known as “Coley’s toxins” [1]. An early and clinically impactful milestone that followed Coley’s work was the introduction of Bacillus Calmette–Guérin (BCG), a live-attenuated *Mycobacterium bovis* strain, for the treatment of non-muscle-invasive bladder cancer (NMIBC). Since its approval in the 1970s, intravesical BCG remains the gold-standard adjuvant immunotherapy for NMIBC and represents one of the most successful examples of microbial use in cancer treatment, achieving durable responses through local inflammation, antigen presentation enhancement, and recruitment of cytotoxic immune cells [2]. Recently, the Argentine therapeutic melanoma vaccine VACCIMEL—composed of four lethally irradiated allogeneic melanoma cell lines—received regulatory approval, and is administered specifically in combination with BCG (and GM-CSF) as adjuvants [3], underscoring not only the resurgence of microbial-based immunotherapy but also the enduring relevance of BCG as a potent immune stimulant in treatment-adjunct settings. Other live-attenuated or genetically engineered bacteria such as *Salmonella*, *Listeria monocytogenes*, *Clostridium*, and *Bifidobacterium* have also been investigated as vectors for cancer-targeted therapy [4]. These microbes can deliver therapeutic molecules, induce immunogenic cell death, and enhance tumor-associated antigen presentation. Moreover, their capacity to reshape the suppressive TME into a proinflammatory one, by inducing pro-inflammatory cytokines and recruiting immune cells, places them as promising agents in both monotherapy and combination immunotherapy strategies. It is important to state that an effective bacterial candidate for anticancer therapy must strike an adequate balance between attenuation and antitumor activity. While wild-type strains are not suitable for clinical use due to their ability to induce severe systemic infections, overly attenuated derivatives may show reduced therapeutic efficacy because they fail to elicit a sufficient immune response for tumor regression [5].

*Salmonella enterica* serovar Typhimurium has been extensively studied as a vector for cancer immunotherapy due to its ability to preferentially colonize tumors, kill tumor cells and stimulate robust immune responses [6]. This is in part because of its capacity to use its flagella to move towards the TME, attracted by the available nutrients [7,8]. In addition, being a facultative anaerobe, *Salmonella* thrives in the typical hypoxic, poorly vascularized microenvironments of solid tumors, a property that has been exploited to design hypoxia-conditional strains such as YB1 that survive and replicate selectively within tumor cores [9]. Therefore, in this way, *Salmonella* acts at different levels: by starving tumor cells through competition for the same nutrients [10] or metabolically disrupting the TME [11]. In addition, *Salmonella* can invade tumor cells within the whole tumor mass (both viable margins and also the necrotic core, inaccessible for other therapies) and directly induce their death.

Beyond preclinical evidence, some attenuated *Salmonella* strains have already been evaluated in clinical settings, supporting the feasibility of exploiting the bacterium as an antitumoral agent. The first-in-human Phase I clinical trial administering intravenous attenuated *S. Typhimurium* VNP20009 to patients with metastatic melanoma demonstrated that treatment was safe at low doses, with occasional tumor colonization observed and evidence of immune activation, although without objective clinical responses [12]. A subsequent study using continuous intravenous infusion of the same strain further confirmed tolerability, setting safety parameters for future interventions [13]. More recently, an oral formulation engineered to express human IL-2 was tested in patients with metastatic gastrointestinal cancer, also showing an acceptable safety profile and preliminary immune engagement, thus reopening interest in clinical development [14]. Together, these trials highlight that *Salmonella*-based therapies are clinically implementable and provide a translational framework for next-generation tumor-targeting strains optimized for enhanced safety, delivery, and antitumor efficacy.

Although *Salmonella* exerts diverse antitumor activities beyond direct induction of tumor cell death, these aspects have been extensively covered elsewhere [15,16] and fall outside the specific scope of this review. The use of *Salmonella* in combination with other therapies has also been reviewed elsewhere [16]. Therefore, in this review we intend to specifically address the different *Salmonella*-induced types of cell death (both in tumor and immune cells in the tumor bed) that, together with other phenomena, explain the preclinical success of cancer immunotherapies involving the use of attenuated *Salmonella*. The main objective of cancer therapies is the elimination of tumor cells, hence, the fact that *Salmonella* is able to induce diverse types of tumor cell death at the same time is highly relevant. The balance between these multiple cell death pathways dictates the final outcome of the therapy, which may favor the development of a tumor-specific effective immune response and therefore, disease eradication. Finally, we briefly discuss the rationale and challenges of manipulating tumor cell death mechanisms.

## 2. *Salmonella*-Induced Cell Death

Among the various contributing factors, *Salmonella*-mediated direct killing stands out as a major mechanism underlying the antitumor effects [17]. Different investigations have shown reduced tumor cell viability upon infection, using viability assays such as MTT [18] and resazurin reduction [19], or death assays as lactate dehydrogenase (LDH) release [20]. However, these approaches, while indicative of cytotoxicity, do not clarify the precise mode of cell death involved. A growing body of evidence indicates that *Salmonella* is capable of activating multiple cell death pathways, with outcomes depending on bacterial strain, host cell type, TME and other factors, as reviewed below. This distinction is biologically relevant, since different types of cell death can trigger distinct downstream immune consequences. Beyond the primary reduction in tumor burden through direct killing, the byproducts and danger signals released during each death process can differentially modulate antitumor immunity. Understanding these distinct mechanisms and their crosstalk is crucial for optimizing *Salmonella*-based cancer therapies. Therefore, in the following sections we provide an overview of each type of programmed cell death associated with *Salmonella* in cancer control, which are schematically represented in Figure 1.

### 2.1. Apoptosis

This type of programmed cell death is one of the most studied in the context of *Salmonella*-mediated cancer therapy (Table 1). It consists of controlled self-destruction of cells, while the cell membrane remains intact and does not induce inflammation [21]. Notably, *S. Typhimurium* has been shown to both induce and modulate apoptosis in tumor cells through various mechanisms. Below, we summarize how *Salmonella* triggers apoptosis in different cells, the key molecular players involved, and how these phenomena contribute to tumor suppression and immunotherapy (Table 1).

The first report of *Salmonella* inducing apoptosis in infected macrophages dates back to 1996, in a work published by Monack et al. [22]. Subsequent work revealed the mechanism: the *Salmonella* pathogenicity island-1 (SPI-1) Type III secretion system (T3SS1) delivers an effector protein, SipB, into host macrophages that is both necessary and sufficient to induce apoptosis [23] (Figure 1). Notably, Caspase-1-mediated cell death is now termed pyroptosis (see Section 2.2 below) but earlier work described it as an apoptotic phenotype. In human macrophage-like THP-1 cells, *Salmonella* infection can also induce rapid DNA fragmentation in the host cells within hours. Valle and Guiney found that >70% of THP-1 cells became TUNEL-positive (indicating DNA breaks) after 4 h of infection, increasing to >90% by 5.5 h [24]. This effect depended on active bacterial infection and, surprisingly, did not require the SPI-1 or SPI-2 secretion systems or the spv virulence genes, but did require the *phoP* regulatory gene. Caspase-3 (an executioner caspase in classical apoptosis) was activated during *Salmonella* infection of THP-1 cells, though Caspase-8/9 (initiators of extrinsic/intrinsic apoptosis) were not, and blocking Caspase-3 did not fully prevent DNA fragmentation. These findings suggest *Salmonella* can initiate an atypical apoptotic program in macrophages, likely through Caspase-1 (pyroptotic) pathways and additional PhoP-regulated factors. The death of infected macrophages via apoptosis/pyroptosis may benefit the pathogen by releasing it from the macrophage, but in the context of cancer therapy, pyroptosis, but not apoptosis, can help expose tumor antigens and promote inflammation in the TME. In contrast with the pro-apoptotic effect seen in macrophages, *Salmonella* can actively suppress apoptosis in intestinal epithelial cells during the early stages of natural infection, ensuring a stable intracellular niche for replication. A key SPI-1 effector, SopB (also known as SigD), has been shown to protect infected epithelial cells from undergoing apoptosis [25]. In this work, Knodler et al. also demonstrated that wild-type *Salmonella* induces sustained Akt activation in epithelial cells in a SopB-dependent manner, and this prevents the normal apoptotic cascade.

Regarding the use of *Salmonella* in cancer therapy, studies in mouse tumor models have shown that systemically delivered *Salmonella* accumulate inside tumors and can trigger tumor cell death. Ganai et al. tracked *S. Typhimurium* in 4T1 mammary carcinoma and found that, within 48 h of intravenous injection, the bacteria had migrated away from well-vascularized tumor edges into the hypoxic tumor core, coinciding with elevated apoptosis in this central tumor region [26]. Hence, this study demonstrated that *Salmonella* not only penetrate deep into tumor tissue but also induce cancer cell apoptosis in vivo, contributing to a temporary stagnation of tumor growth.

Engineering *Salmonella* can further enhance this pro-apoptotic oncolytic effect. Kasinskas and Forbes showed that deleting the bacterial ribose/galactose chemoreceptor (which normally guides *Salmonella* toward certain nutrients) causes the bacteria to linger in tumor quiescent zones, thereby intensifying their apoptotic impact on tumor cells [8]. The chemotaxis-deficient strain accumulated more in poorly perfused tumor regions and induced higher Caspase-3 activity (i.e., more apoptosis) in the tumor compared to wild-type *Salmonella*. Thus, *Salmonella*’s inherent tumor tropism can be tuned to maximize direct tumor cell killing via apoptosis. Recent research confirms that *Salmonella*-mediated tumor cell apoptosis is a key mechanism of tumor suppression. Zhang et al. employed an attenuated *S. Typhimurium* strain in a “L-form” (cell wall-deficient state) to treat murine ovarian cancer, and observed a robust induction of apoptosis in the ovarian tumor cells, alongside with inhibited tumor growth [27]. Tumors from treated mice showed significantly more TUNEL-positive (apoptotic) cancer cells than controls, and the therapy reduced expression of pro-tumor proteins, like galectin-9 and MMP9, which are associated with tumor proliferation and metastasis. This confirms that *Salmonella* can directly trigger intrinsic cell death pathways in cancer cells, contributing to oncolysis. In some studies, *Salmonella* infection has also been noted to damage the tumor vasculature and endothelium, partly through inducing apoptosis of endothelial cells, which ends up in an anti-angiogenesis effect that feeds a positive loop [28].

Several bacterial and host factors underlie *Salmonella*-mediated apoptosis. Among them, SipB is a key *Salmonella* effector during macrophage infection that binds and activates Caspase-1. At the time, this was described as an “inflammatory apoptotic death” of the phagocyte; however, it is now recognized as pyroptosis [23]. SipA, another SPI-1 effector, can activate Caspase-3 in host cells, linking *Salmonella* infection to classical apoptosis execution pathways [29]. In infected macrophages, Caspase-1 activation by SipB can also indirectly activate Caspase-3 downstream or induce other apoptotic signals, although the exact signaling crosstalk is complex. In the tumor context, host cytokines like TNF induced by *Salmonella* are also key pro-apoptotic mediators. *Salmonella*’s lipopolysaccharide (LPS) triggers TNF secretion by macrophages and monocytes, which can cause apoptotic death of tumor endothelial cells and tumor cells, contributing to tumor oncolysis as stated above [6,30].

All in all, harnessing *Salmonella*-mediated apoptosis—through careful genetic attenuation and targeting—may offer a unique biotherapeutic route for cancer treatment. Ongoing research and clinical trials will reveal how this strategy can be optimally and safely integrated into cancer immunotherapy, potentially providing a powerful adjunct or alternative to conventional treatments.

**Table 1 biomedicines-14-00012-t001:** Programmed cell death events associated with *Salmonella* antitumor effect in in vivo preclinical models.

Tumor Type	*Salmonella* Strain and Route of Administration	Type of Induced Cell Death	Refs.
Breast cancer (MARY-X human xenograft)	A1-R (i.v.)	Apoptosis *	[31]
Breast carcinoma (4T1)	Engineered VNP20009 (i.v.)	Apoptosis	[26]
Colon adenocarcinoma (CT26)	StΔppGpp-lux (i.v.)	Apoptosis/Pyroptosis *	[32]
Colon adenocarcinoma (CT26)	Engineered SGKS1004 (i.v./i.t.) (*S. enterica* serovar Gallinarum)	Autophagy	[33]
Colon adenocarcinoma (MC38)	ΔppGpp (i.v.)	Pyroptosis *	[34]
Fibrosarcoma (HT1080)	A1-R (i.g.)	Autophagy	[35]
Glioma (U87)	YB1 (i.v.)	Ferroptosis	[36]
Laryngeal cancer (Hep-2)	Engineered LH430 (i.v.)	Apoptosis	[37]
Non-small cell lung carcinoma (A549)	Engineered VNP20009 (i.v.)	Apoptosis	[38]
Ovarian cancer (ID8)	L-form VNP20009 (route not specified)	Apoptosis	[27]
Prostate adenocarcinoma (PC-3)	A1-R (i.v.)	Apoptosis *	[39]
Skin melanoma (B16F1)	LVR01 (i.t.)	Apoptosis */ICD/Pyroptosis	[40]
Skin melanoma (B16F10)	VNP20009 (i.v.)	Apoptosis	[9,41]
Skin melanoma (B16F10)	VNP20009 (i.p.)	Apoptosis/Autophagy	[42]
Skin melanoma (B16F10)	Engineered VNP20009 (i.g.)	Apoptosis	[43]
Skin melanoma (B16F10)	Engineered VNP20009 (i.p.)	Apoptosis	[44]
Skin melanoma (B16F10)	Engineered 14028 (i.t.)	Apoptosis	[45]

All strains were *S. enterica* serovar Typhimurium, unless specifically stated otherwise. * indicates indirect evidence of particular cell death, through a mechanism not directly named in the article. “Engineered” depicts bacterial modifications to enhance/delete/add certain molecules. Abbreviations: ICD, immunogenic cell death; i.v., intravenous; i.t., intratumoral; i.g., intragastric (gavage); i.p., intraperitoneal.

### 2.2. Pyroptosis

Pyroptosis is a lytic and highly inflammatory form of programmed cell death driven by the gasdermin family of pore-forming proteins [46]. It is characterized by cell swelling, membrane permeabilization, and release of pro-inflammatory intracellular contents, in contrast to the silent dismantling of apoptosis. Firstly discovered as a defense mechanism against intracellular bacteria, pyroptosis occurs when cytosolic pattern recognition receptors engage their ligand and trigger the assembly of cytosolic multiprotein complexes called inflammasomes, which activate inflammatory caspases (such as Caspase-1, -4/5 in humans, or -11 in mice) in response to microbial or danger signals [47,48]. These caspases cleave the inhibitory C-terminal domain off gasdermin (GSDM) proteins, unleashing the cytotoxic N-terminal domain to oligomerize in the plasma membrane and form large pores, thereby disrupting ionic homeostasis and causing cell lysis. This gasdermin-mediated pore formation is the executioner step of pyroptosis and is essential for its downstream effects [49,50]. Indeed, cells lacking GSDMD, the substrate of Caspase-1/4/5/11, are resistant to pyroptosis and fail to secrete the potent cytokines interleukin-1β (IL-1β) and IL-18 upon infection [49]. Cleavage of GSDMD by inflammatory caspases releases its N-terminal fragment, which not only perforates the membrane to induce pyroptotic death but also can prompt assembly of the NLRP3 inflammasome, amplifying IL-1β maturation in a positive feedback loop [49] (Figure 1).

*Salmonella* infection is a quintessential trigger of pyroptosis. It delivers pathogen-associated molecular patterns (PAMPs) into the host cell cytosol, activating multiple inflammasome pathways. In macrophages, *Salmonella*’s flagellin and type III secretion system rod proteins are recognized by NAIP/NLRC4 inflammasomes, while additional stress signals from infection (for example, phagosomal damage or bacterial toxins) can engage the NLRP3 inflammasome [48]. Both NLRC4 and NLRP3 converge on Caspase-1 activation and were found to play partially redundant roles in host defense against *Salmonella*: mice deficient in both of these inflammasome sensors are markedly more susceptible to infection [48]. *Salmonella* can also induce “non-canonical” inflammasome activation via Caspase-11 in mice (Caspase-4/5 in humans) when its LPS enters the cytosol. Consequently, *Salmonella*-infected macrophages rapidly activate Caspase-1 and Caspase-11, leading to GSDMD cleavage and pyroptotic cell death [49]. Along with directly killing the pathogen’s replicative niche, pyroptotic death of *Salmonella*-infected cells results in the maturation and release of IL-1β and IL-18 (Figure 1), which act on surrounding cells to induce fever, vascular permeability and the recruitment of immune effector cells [48,49]. Pyroptotic cells also release alarmins such as ATP, HMGB1, and inflammasome complexes themselves; for example, oligomerized ASC “specks” are expelled from pyroptotic cells and can be phagocytosed by bystander macrophages to propagate Caspase-1 activation and cytokine release in the local environment [51]. Through these inflammatory feedback mechanisms, *Salmonella*-induced pyroptosis serves to alert and mobilize the immune system, effectively linking innate pathogen sensing to a potent inflammatory response that contains infection. Given its powerful immunostimulatory nature, pyroptosis has consequences in the context of cancer and is being explored as a tool in cancer therapy. Notably, *Salmonella*-induced pyroptosis has been identified as a key mechanism by which these bacteria can provoke an anticancer immune response. In a mouse colon adenocarcinoma model, intratumoral administration of an attenuated *S. Typhimurium* elicited significantly elevated IL-1β levels in the TME and resulted in suppressed tumor growth, an effect that was dependent on inflammasome signaling. In addition, *Salmonella*-treated tumors showed upregulation of inflammasome pathway components (including NLRC4, NLRP3, and the P2X7 purinergic receptor), indicating that the bacterium activates innate immune sensors within the tumor. Different mechanistic studies revealed that *Salmonella* triggers pyroptosis predominantly in tumor-infiltrating macrophages: the bacteria directly infect these phagocytes and activate their inflammasomes, and additionally, *Salmonella*-mediated damage to nearby tumor cells releases Danger-Associated Molecular Pattern (DAMPs) (e.g., extracellular ATP) that further engage macrophage NLRP3 [34,40]. The result of this coordinated activation is pyroptotic death of some tumor-associated macrophages and robust IL-1β release in the tumor milieu. Actually, it has been shown that levels of IL-1β (and also TNF) are markedly increased in tumors colonized by *Salmonella*, but when they restart growth, cytokine levels return to normal. In addition, local administration of IL-1β along with *Salmonella* therapy further prolonged tumor suppression [32]. IL-1β is a potent proinflammatory cytokine that, along with IL-18 (also produced during inflammasome activation), can promote the recruitment and activation of lymphocytes and other immune cells in the tumor [48]. Thus, *Salmonella*-induced pyroptosis turns the immunosuppressive TME into an acutely inflamed site, which is conducive to antitumor immune activity. Consistent with this, *Salmonella* therapy failed to inhibit tumor growth in mice lacking Caspase-1/11, despite having normal baseline levels of IL-1β. This indicates that the process of pyroptosis (Caspase-1/11-mediated cell death)—and not merely the presence of IL-1β—is required to achieve tumor suppression, likely because pyroptosis amplifies local inflammation and releases other immunostimulatory factors (such as IL-18, HMGB1, and cleaved gasdermin-D fragments) that together drive a productive antitumor response [40]. In support of pyroptosis being an immunogenic cell death, a study showed that *Salmonella*-infected tumors showed hallmarks of immunogenicity: dying tumor cells exposed calreticulin on their surface (an “eat me” signal for dendritic cells) and released HMGB1, indicating that pyroptosis was occurring in tumor cells as well as in macrophages. The involvement of tumor cell pyroptosis in *Salmonella* therapy is further suggested by observations that *Salmonella*-infected melanoma cells upregulate Caspase-11 and gasdermin-D and undergo membrane permeabilization. Notably, the efficacy of *Salmonella* against tumors was also abolished in NLRP3-deficient mice and upon macrophage depletion [40], underscoring that both the sensor (NLRP3 inflammasome) and the effector cells (macrophages) driving pyroptosis are essential for this form of bacteria-mediated cancer therapy. Together, these findings illustrate that *Salmonella*-induced pyroptosis within the TME activates innate immunity in situ and can spark downstream adaptive immune responses that curb tumor progression. Pyroptosis thereby serves as a critical link between *Salmonella* infection and cancer treatment by acting as an engine for inflammation-driven antitumor immunity. Pyroptotic cell death releases a wealth of inflammatory mediators and tumor antigens that help the immune system recognize and attack cancer. For example, the GSDMD pores not only allow IL-1β and IL-18 secretion, but also permit the release of tumor-associated antigens and DAMPs from dying tumor cells, which can be taken up by dendritic cells to prime tumor-specific T cells [52,53]. The concomitant release of IL-18 can drive IFN-γ production by NK and T cells, while IL-1β and HMGB1 recruit and activate myeloid cells, altogether creating a highly immunostimulatory TME [52,53]. Accordingly, inducing pyroptosis in even a fraction of tumor cells can trigger a systemic anticancer immune response. Recent studies have demonstrated that pyroptotic death in as few as ~10–15% of 4T1 tumor cells is sufficient to generate an inflammatory cascade that recruits cytotoxic lymphocytes to eradicate the remaining tumor [54], and that *Salmonella*-mediated antitumor effect disappears if T cells or NK cells are absent [55]. Pyroptosis within these tumors has been shown to increase infiltration of CD8^+^ T cells, CD4^+^ T cells, and NK cells, while reducing immunosuppressive cell types such as myeloid-derived suppressor cells, and to drive macrophages toward a pro-inflammatory M1 phenotype. This shift from an “immune-cold” tumor (poorly infiltrated by T cells) to an “immune-hot” tumor (rich in T cell infiltration) is precisely what is needed to improve responsiveness to immunotherapies. Indeed, pyroptosis is now recognized as a form of immunogenic cell death that can convert the TME from suppressive to highly inflammatory, thereby potentiating therapies like immune checkpoint blockade (revised in [53]). In preclinical models of colon carcinoma and melanoma, combination strategies that pair checkpoint inhibitors with pyroptosis-inducing agents lead to superior tumor control, suggesting a synergistic effect where pyroptosis fuels antitumor T cell responses [56]. Accumulating evidence indicates that robust activation of pyroptosis pathways tends to halt tumor growth by activating immune surveillance, whereas defects or suppression of pyroptosis can allow tumors to evade immune detection. For instance, it has been reported that NLRP3 inflammasome components deficiency significantly correlates with advanced hepatocellular carcinoma stage [57]. Many cancers downregulate key pyroptosis executors, particularly different gasdermins. In gastric cancers GSDMA, C and D have been shown to downregulate their expression and have even been postulated as tumor suppressors [58,59,60], although in those studies pyroptosis was not yet described. In addition GSDMD is also downregulated in colorectal carcinoma (CRC) [61]. For all the above-mentioned situations, restoring or introducing the capacity for pyroptosis in such tumors can tip the balance toward inflammatory cell death and tumor rejection. In that regard, *Salmonella* has shown to induce GSDMD upregulation in B16F1 melanoma cells [40]. In that report, a single intratumoral injection with *Salmonella* also induced augmentation of transcription of *NLRP3* and *NLRC4* inflammasome receptor genes in tumor cells three days post treatment, as well as executors Caspase-1 and IL1β, the latter both at mRNA and protein levels, leading to delay in tumor growth and prolonged survival of melanoma bearing mice.

Pyroptosis has been described as a double-edged sword: while its chronic activation can drive inflammation that may promote tumor development, its acute induction during therapy can exert potent anticancer effects [62]. Therefore, harnessing this form of inflammatory cell death is a promising strategy in cancer treatment. *Salmonella*-induced pyroptosis exemplifies this strategy: by causing targeted immune cell death in the tumor and flooding the area with inflammatory signals, it effectively turns the patient’s own immune system into a weapon against the cancer. Hence, the goal is to build on these insights—for example, engineering *Salmonella* or other vectors to trigger tumor-localized pyroptosis more efficiently—to overcome suppressive TME and amplify antitumor immunity, offering a potent adjunct or alternative to conventional cancer treatments that would improve cancer immunotherapy outcomes.

### 2.3. Autophagy

Autophagy is a lysosome-dependent form of cell death that occurs at a low rate under homeostatic conditions but it is amplified in response to cellular stresses such as nutrient deprivation, hypoxia, energy depletion or infection [63]. It is a conserved and essential cellular process through which dysfunctional organelles and cytoplasmatic components are degraded and recycled. Autophagy also targets intracellular bacteria for lysosomal degradation (a process known as xenophagy), thus playing a crucial role in innate immunity [64,65]. Mechanistically, autophagy is a catabolic process in which protein aggregates, invading pathogens and damaged organelles are sequestered into LC3-positive double-membrane vesicles called autophagosomes. These vesicles then fuse with lysosomes, where their cargo is degraded by lysosomal hydrolases [66] (Figure 1).

Host cells can trigger autophagy upon *Salmonella*-infection by directly recognizing cytosolic bacteria and by detecting damaged *Salmonella*-containing vacuoles (SCVs). Following the active invasion, *S. Typhimurium* typically resides within SCVs, whose integrity can be compromised by the activity of the T3SS1 [67]. When bacterial components are exposed to the cytosol, they induce host ubiquitination pathways and the recruitment of autophagy adaptor proteins that bind ubiquitinated bacteria/damaged SCV and LC3, facilitating their sequestration into autophagosomes [68]. In autophagy-deficient cells (e.g., *Atg5*^−^/^−^), intracellular replication of *Salmonella* is enhanced, highlighting the role of autophagy in restricting bacterial proliferation and maintaining cytosolic integrity. *Salmonella* can also trigger autophagy through nutrient-sensing pathways. The serine/threonine kinase mTOR (mammalian Target of Rapamycin) acts as a central metabolic checkpoint that integrates signals related to nutrient availability, growth factors, and cellular energy status to regulate processes such as protein synthesis, cell growth, and autophagy. Under nutrient-rich conditions, mTOR activity suppresses autophagy. However, during active invasion, the insertion of the T3SS1 translocation apparatus into the host plasma membrane forms pores that cause acute leakage of amino acids from the cytosol. This sudden nutrient depletion triggers a starvation-like response, altering the subcellular localization of mTOR and reducing its activity. As a result, autophagy is rapidly initiated as part of the host defense. Notably, this autophagic response is transient, peaking at approximately 1 h post-infection and declining by 3 h, which coincides with the downregulation of T3SS1 and the upregulation of T3SS2. This transition may restore membrane integrity and normalize amino acid levels, contributing to the resolution of the autophagic cascade [69,70].

In addition to restricting bacterial replication, autophagy plays a crucial role in modulating inflammatory responses. Inflammatory stimuli can activate autophagy to limit IL-1β production by targeting ubiquitinated inflammasome components for lysosomal degradation [71]. Recently, the adaptor protein p62 was shown to inhibit IL-1β release in macrophages infected with *Salmonella Typhimurium* by promoting the autophagic degradation of inflammasome complexes [72]. These findings highlight the role of selective autophagy in disrupting inflammatory signaling by modulating other mechanisms such as pyroptosis.

Autophagy has a dual role in tumor development; it can promote tumor progression or regression depending on different features such as the type of cancer and the stage of the disease, the genetic background, and the TME characteristics, among others [73,74,75]. This opposing role is well illustrated by a study from Rao et al., who used a murine model of lung cancer to compare autophagy-competent and autophagy-deficient (*Atg5*-deleted) tumors. They found that loss of autophagy accelerated early tumor initiation but reduces progression from adenomas to adenocarcinomas. This context-dependent behavior highlights the need for caution when considering autophagy modulation as a therapeutic strategy in cancer [76]. The stage of cancer appears to be particularly critical: while autophagy is required for cancer immunosurveillance in early stages, once tumors are established, increased autophagy may favor tumor cell survival and growth [77,78] (reviewed in [79]).

In addition to its role in tumor cells, autophagy is also crucial in influencing the behavior of immune cells within the TME. For instance, pharmacological inhibition of autophagy with chloroquine in tumor-associated macrophages has been shown to promote a shift from an immunosuppressive M2 phenotype to a proinflammatory M1-like phenotype. This change leads to a reduction in immunosuppressive T cell populations and an enhancement of antitumor T cell immunity in mouse models [80]. Similarly, inhibition of autophagy in CD8^+^ T cells enhanced tumor rejection by promoting a more effector memory-like phenotype associated with increased production of IFN-γ and TNF. These effects were linked to enhanced glucose metabolism and epigenetic remodeling in T cells [81].

These findings highlight the potential of targeting autophagy to modulate immune responses within tumors. In this regard, it has been demonstrated that *Salmonella* can induce autophagy in tumor cells by downregulating the AKT/mTOR pathway [82,83]. *Salmonella*-induced autophagy contributes to the suppression of indoleamine 2,3-dioxygenase 1 (IDO), a key immunosuppressive molecule in the tumor microenvironment, thereby hindering tumor immune tolerance and promoting antitumor immunity [84]. Furthermore, combining *Salmonella*-based therapy with pharmacological inhibitors of autophagy, such as chloroquine, has been shown to enhance antitumor effects. This combinatorial approach improves bacterial tumor targeting, increases tumor cell death, and boosts antitumor immune responses [42,85]. For instance, the use of *Salmonella* strains VNP20009 or A1-R in combination with chloroquine resulted in synergistic tumor eradication, both in vitro and in vivo, particularly in fibrosarcoma and other solid tumor models [35,85]. Similarly, the co-administration of chloroquine with *S. Gallinarum* expressing arginine deaminase (*PelB*-ADI) enhanced tumor growth inhibition by simultaneously targeting tumor metabolism and disrupting autophagic flux [33].

Altogether, the complex interplay between autophagy, immune modulation, and tumor progression highlights the therapeutic potential of strategically modulating autophagy within *Salmonella*-based approaches to boost antitumor immunity.

### 2.4. Emerging Cell Death Pathways Potentially Involved in Salmonella-Based Cancer Immunotherapy

While apoptosis, pyroptosis, and autophagy are the best-characterized forms of cell death associated with *Salmonella*-based cancer immunotherapy, other regulated cell death pathways may also influence therapeutic outcomes. Although experimental evidence directly linking them to cancer treatment efficacy remains limited, the following mechanisms could play relevant roles in shaping the immune and metabolic landscape of the TME.

#### 2.4.1. Necroptosis

Necroptosis is a form of regulated necrosis mediated by death receptors and typically triggered by Caspase-8 inhibition. Although it represents another type of inflammatory programmed cell death, it differs from pyroptosis in several aspects. One major distinction lies in the key signaling molecules involved, namely receptor-interacting protein kinase 1 (RIPK1), receptor-interacting protein kinase 3 (RIPK3) and their downstream substrate, mixed lineage kinase domain-like (MLKL). Upon activation, RIPK1 and RIPK3 interact to form the necrosome, initiating a phosphorilation cascade that recruits MLKL, ultimately leading to necroptotic cell death [86]. Activated MLKL oligomerizes and translocates to the plasma membrane, forming selective pores that cause calcium influx and culminate in explosive, lytic cell death [87]. To date, no direct association between *Salmonella* cancer immunotherapy and necroptosis has been established. However, Hancz et al. have demonstrated in Jurkat T cells that *Salmonella* can trigger RIPK1 (formerly depicted as RIP1)-dependent cell death through flagellin, while the apoptotic pathway remains unaffected and in the absence of caspases, also known as necroptosis [88]. Also, *Salmonella* infection leads to downregulation of Caspase-8 in infected macrophages and induces necroptosis through type I interferon in the presence of pancaspase inhibition [89,90]. In addition, it is known that intratumoral necroptosis potentiates antitumor immunity [91]. Thus, in a Caspase-8 inhibition scenario induced by *Salmonella*, the resulting necroptosis may lead to a better treatment outcome, pinpointing Caspase-8 as a good target for *Salmonella*-based cancer treatment. Nevertheless, since tumor expression of Caspase-8 varies within cancer types [92], this aspect needs to be further studied in each particular situation.

#### 2.4.2. PANoptosis

As knowledge of programmed cell death has expanded, a recent unifying concept has emerged: PANoptosis, the coordinated execution of pyroptosis, apoptosis and necroptosis through a single regulatory node. This regulation allows for circumventing inhibition of individual death pathways [93], which may represent a tumor cell strategy to avoid elimination. Caspase-8 sits at the hub of this “death triad,” acting as a molecular switch that can license or suppress each pathway depending on context [94,95], and all these processes are actually controlled by the same master regulators (ZBP1 and TAK1) [96]. A work in murine bone marrow macrophages showed that *Salmonella* activates the NAIP/NLRC4 inflammasome, recruiting Caspase-1/11 and Caspase-8 into a multimeric “PANoptosome”. Only cells lacking all four enzymes (Caspase-1, -11, -8 and RIPK3) escape death, confirming simultaneous engagement of all three arms of PANoptosis, whereas single knockouts do not [93]. This fail-safe design thwarts *Salmonella*’s attempts to block individual pathways and ensures robust cytokine release (IL-1β, IL-18, HMGB1) together with lytic membrane rupture of infected host cells. Since the resulting microenvironment is flooded with danger cues, it constitutes an ideal setting for mounting antitumor immunity. Hence, the use of attenuated *Salmonella* as immunotherapy exploits the same circuitry inside tumors. An example of the combined forces is the evidence that *Salmonella* treatment in colon carcinoma model promoted tumor regression associated with increased IL-1β and TNF, Caspase-1 activation, and NLRC4 inflammasome signaling [32], even when by that time the term PANoptosis term was not coined yet. All in all, PANoptotic cell death converts “cold” tumors into inflamed foci rich in antigen, DAMPs and type-1 cytokines—conditions that foster dendritic-cell priming and cytotoxic-T/NK-cell recruitment. Harnessing or mimicking this integrated death program—rather than stimulating a single pathway—may offer a powerful route to next-generation cancer immunotherapies, although the implications of PANoptosis on cancer immunotherapies are not fully explored nor understood.

#### 2.4.3. ETosis

Other less common ways of cell death have been reported by the use of *Salmonella*, as is the case of ETosis, in which the cell releases its contents into the extracellular space forming DNA nets which are termed “extracellular traps” [97]. Particularly regarding METosis, the ETosis of macrophages, it has been reported that *Salmonella* infection steadily induces the release of Macrophage Extracellular Traps (METs) with the concomitant cell death by METosis in less than an hour [98]. Nevertheless, its role in cancer was not assessed. In this regard, the group of Cools-Lartigue et al. has found that a close relative of METs, which are the Neutrophil Extracellular Traps (NETs), may favor metastasis development by sequestering tumor cells and releasing them in distant places [99,100]. On the other hand, NETs have been reported to boost chemotherapy against CRC through cathepsin G-mediated translocation of cytosolic BAX to the mitochondrial membrane. There, BAX and BAK oligomerize to form pores that mediate cytochrome c release, promoting apoptosis [101]. To the best of our knowledge, up to date no mechanistic link between METs and cancer has been described. However, since extracellular DNA acts as a DAMP that activates inflammasome [102,103], it is plausible that its detection may induce an inflammatory response, that together with antigen exposure caused by tumor cell death by other pathways (e.g., *Salmonella*-induced pyroptosis), may generate a specific antitumor response. Further studies are needed to elucidate the outcome of the occurrence of extracellular traps release in a tumoral context. However, there are reports indicating that protein components of these traps, such as myeloperoxidase and elastase, exhibit antitumor effects in vivo, both locally and even at distant metastatic sites [104,105]. Moreover, elastase has been shown to selectively kill tumor cells without affecting non-tumor cells [105]. In addition, NETs have been found to correlate positively with survival in patients with high-grade ovarian cancer, through the induction of NADPH oxidase 2-independent NETosis [106] and in the case of CRC, increased NET levels in tumors were also associated with longer progression-free survival [101].

Interestingly, the GSDMD pores required for pyroptosis may also be used to release the extracellular traps, so both processes have proven to be connected by this molecule [107,108], even when there are also reports regarding NETosis independent of GSDMD [109]. Therefore, a systematic simultaneous assessment of pyroptosis and ETosis, their interaction and contribution to *Salmonella*-based cancer immunotherapy is still required.

It is also worth mentioning that M1 macrophages, unlike their M2 counterparts, are capable of releasing METs as part of their antimicrobial and proinflammatory response [110]. As stated in Section 2.3, inhibition of autophagy favors M1 polarization [80], and this effect may partly result from impaired clearance of inflammasome components, which enhances Caspase-1 activation and IL-1β release, thereby amplifying inflammatory cell death [111,112]. We have previously reported that soon after intratumoral injection, *Salmonella* induces pyroptosis and promotes a transient shift from M2 to M1 macrophages that corresponds to antitumor effect, but this profile is then lost [40]. In this context, modulation of autophagy may represent a strategy to sustain or potentiate the M1 phenotype and the proinflammatory responses elicited by *Salmonella* within the TME, perpetuating its therapeutic effects. Interestingly, inhibition of autophagy has been shown to prevent intracellular chromatin decondensation, a key step for NETosis, leading instead to cell death resembling apoptosis [113]. This finding suggests that apoptosis might act as a backup program when ETosis pathways are impaired, highlighting the complex interplay between different cell death mechanisms.

#### 2.4.4. Ferroptosis

Ferroptosis is a newly characterized iron-dependent form of regulated cell death triggered by disruption of cellular redox homeostasis, leading to the accumulation of lipid peroxides and oxidative damage [114]. The only report regarding *Salmonella*-induced ferroptosis as part of an immunotherapy is on glioma cells, showing that *Salmonella* YB1 infection disrupts the GPX4–glutathione redox axis, leading to mitochondrial damage and tumor growth suppression in vivo, which can be reversed by the use of the ferroptosis inhibitor Fer-1 [36]. It has been reported that therapy-resistant cancer cells, particularly those of the mesenchymal state and prone to metastasis, are profoundly vulnerable to ferroptosis. Hence, modulation of ferroptosis, holds great potential for the treatment of cancers (revised in [115]). To date, the contribution of ferroptosis to *Salmonella*-mediated antitumor activity can be directly attributed to tumor mass cytoreduction resulting from cancer cell death. However, the consequences of ferroptosis activation in the TME should be addressed to help understanding its role in tumor suppression and immune surveillance within *Salmonella*-based cancer therapies.

## 3. Challenges Facing *Salmonella*-Induced Cell Death as Antitumor Strategy

The most straightforward approach to inducing cancer cell death was traditionally the use of conventional chemotherapy or radiotherapy. However, in the current era of immunotherapy, the most evident strategies are now immune checkpoint inhibitors and chimeric antigen receptor (CAR) T cell therapies. Although immune checkpoint inhibitors have shown remarkable success in a subset of patients, a significant proportion of treated individuals exhibit partial or no response, and their use remains limited by immune-related adverse events and high costs. On the other hand, CAR T cell therapy, while highly effective in hematological malignancies, faces considerable challenges in solid tumors, including limited efficacy, severe toxicity, and complex logistics associated with the use of autologous cells. In this context, the use of *Salmonella* as a tumor-targeting agent emerges as an innovative and potentially more cost-effective strategy to induce tumor cell death. Nevertheless, this approach is not exempt from challenges. One of the main difficulties lies in the synergy, crosstalk, and compensatory interactions between molecules that were previously thought to act exclusively within their representative cell death pathways. For instance, intracellular molecules released during one form of cell death (e.g., necroptosis) can be sensed by pattern recognition receptors such as NLRP3—which is notably promiscuous in detecting damage—thereby triggering pyroptosis. Regarding PANoptosis, it has been described that the PANoptosome complex consists of Caspase-8, ASC, RIPK1, RIPK3, NLRP3, and ZBP1, which directly interact with each other [116]. Because these molecules can participate in multiple cell death pathways, it becomes difficult to delineate the specific mechanisms induced by *Salmonella*. In addition, evidence suggests an interconnection between pyroptosis, apoptosis, and necroptosis during *S. Typhimurium* infection [93], and while deletion of molecules such as Caspases 1, 8 and 11 along with RIPK3 protects macrophages from cell death, individual deletions do not fully prevent it, highlighting the challenge of fine-tuning the system [117]. Furthermore, gasdermins, originally characterized for their role in pyroptosis, are now increasingly recognized as effectors in apoptosis as well. Certain members of this family, such as GSDME, directly contribute to mitochondrial permeabilization and the release of apoptotic factors, thus bridging pyroptosis and apoptosis [118]. This functional interplay may also explain why cells often display overlapping features of different death pathways.

Furthermore, it is essential to consider both species-specific and cell-type variations. For instance, murine keratinocytes lack NLRP1 expression and are hence unable to respond to NLRP1 activators, while humans constitutively express this receptor in barrier cells [119], highlighting significant interspecies differences in innate immune signaling pathways. Additionally, cell-type-specific responses further complicate the interpretation scenario as, for instance, myeloid and epithelial cells may engage distinct immune pathways upon sensing of infection [120], and many different kinds of cells coexist in the TME at the same time. In the case of ETosis, the existence of distinct subtypes or compositions depending on the cell and the initial trigger also remains a source of heterogeneity that has to be considered.

Although the focus of this review is on myeloid populations due to their abundance and functional relevance within the TME, it is important to acknowledge that lymphocytes may also undergo *Salmonella*-induced cell death through different mechanisms. However, the current evidence specifically addressing these pathways in lymphoid cells during *Salmonella*-based cancer therapies remains limited. For instance, a report suggests that *Salmonella Typhi* can induce apoptosis in lymphoma-derived T cells in vitro and contributes to tumor regression in murine T cell lymphoma models [121]. However, in this case, lymphocytes are not other than tumor cells. Other studies show that *Salmonella* inhibits both pyroptosis [122,123] and autophagy [124] in B cells to persist, but those reports do not address bacteria as cancer therapy. This knowledge gap represents an opportunity for future investigation to clarify whether and how distinct lymphocyte subsets undergo programmed cell death following bacterial targeting in the TME.

Finally, as described in Section 2.3, considering cell death modulation as a therapeutic strategy in cancer also has to take into account the stage of cancer [125,126].

## 4. Conclusions

In this review, we have discussed the different types of cell death triggered by *Salmonella* that may contribute to tumor shrinkage and, ultimately, tumor eradication. We also summarized the mechanisms by which these processes can be modulated, as well as the challenges associated with their regulation—largely arising from the fact that cellular context critically determines whether induction or inhibition of specific death pathways is the most beneficial strategy. All things considered, the ability of *Salmonella* to induce tumor cell death stands as a cornerstone of its oncolytic and immunotherapeutic potential. Through both direct and indirect mechanisms, this approach not only eliminates tumor cells and release tumor antigens, thereby priming antitumor immunity, but also reshapes the TME by promoting inflammation and immune cell infiltration, effectively converting immunosuppressive “cold” tumors into immunologically active “hot” tumors. Importantly, many antitumor agents previously thought to act primarily through one mechanism are now recognized to act through another or a combination of them or to mutually regulate. This paradigm shift underscores the need to reassess the mechanisms underlying *Salmonella*-based immunotherapies and to exploit the full spectrum of programmed cell death modalities they can trigger. Understanding and harnessing these processes is essential to optimize therapeutic outcomes and to advance the rational design and clinical translation of *Salmonella*-based cancer immunotherapies.

## Figures and Tables

**Figure 1 biomedicines-14-00012-f001:**
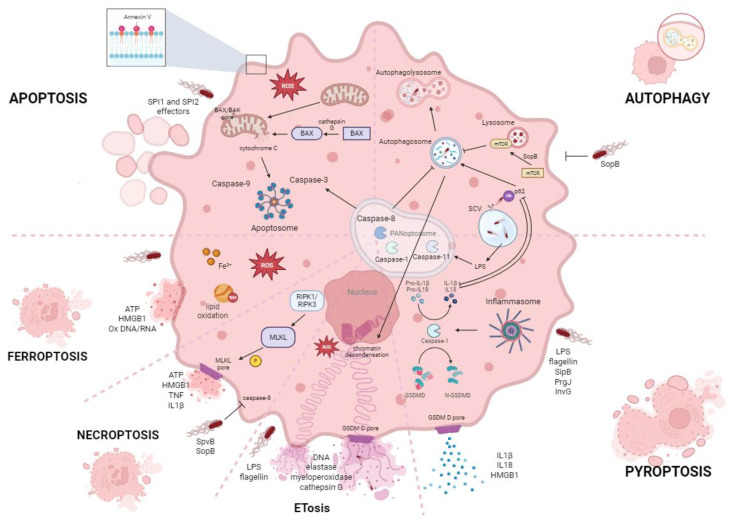
Schematic representation of *Salmonella*-induced cell death pathways, and their coordination and crosstalk. For the sake of clarity, only mouse protein names have been used (some may differ in humans) and several molecules have been omitted. BAX: Bcl-2-associated X protein; GSDM D: gasdermin D, HMGB1: high mobility group box 1, IL: interleukin, LPS: lipopolysaccharide, mTOR: mammalian target of rapamycin, MLKL: mixed lineage kinase domain-like, RIPK: receptor-interacting protein kinase, ROS: Reactive oxygen species, SCV: *Salmonella*-containing vacuole, SPI: *Salmonella* Pathogenicity Island, TNF: tumor necrosis factor. Created in BioRender. Mónaco, A. (2025) https://BioRender.com/w25p0gr (accessed on 7 November 2025).

## Data Availability

No new data were created or analyzed in this study.

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
