# Peer review of "Salmonella-Induced Cell Death in Cancer Immunotherapy: What Lies Beneath?"

_biomedicines, 2025, doi:10.3390/biomedicines14010012_

Round 1

Reviewer 1 Report

Comments and Suggestions for Authors

In the introduction to the manuscript (lines 25-35), the authors make a very brief historical account of the use of microorganisms in cancer therapy. I believe that at least one mention should be included of Bacillus Calmette-Guérin, which has had and has an important place in immunotherapy for cancer.

I suggest that Table 1 be organized into groups by cancer types.

More recent articles should be included. By way of example, I suggest 1. Exp Biol Med (Maywood). 2024 Jun 21;249:10081. DOI: 10.3389/EBM.2024.10081. 

Author Response

The authors thank the reviewer for taking the time to review this manuscript. Please find the detailed responses below and the corresponding revisions highlighted in the re-submitted manuscript.

We have added a paragraph mentioning BCG use for bladder cancer therapy, as well as a recent approval for melanoma patients in Argentina in combination with an irradiated whole-cell vaccine. Lines 33-44 now read “An early and clinically impactful milestone that followed Coley’s work was the introduction of Bacillus Calmette–Guérin (BCG), a live-attenuated Mycobacterium bovis strain, for the treatment of non-muscle invasive bladder cancer (NMIBC). Since its approval in the 1970s, intravesical BCG remains the gold-standard adjuvant immunotherapy for NMIBC and represents one of the most successful examples of microbial use in cancer treatment, achieving durable responses through local inflammation, antigen presentation enhancement, and recruitment of cytotoxic immune cells [2]. Recently, the Argentine therapeutic melanoma vaccine VACCIMEL — composed of four lethally irradiated allogeneic melanoma cell lines — received regulatory approval, and is administered specifically in combination with BCG (and GM-CSF) as adjuvants [3], underscoring not only the resurgence of microbial-based immunotherapy but also the enduring relevance of BCG as a potent immune stimulant in treatment-adjunct settings.”.

Regarding Table 1, we have now reorganized it in alphabetical order of cancer types and also added the provided reference.

Reviewer 2 Report

Comments and Suggestions for Authors

The authors focus on the mechanisms of tumor cell death induced by Salmonella-based cancer therapy, which has recently attracted attention for its ability to modulate the tumor microenvironment (TME). The review is well-structured and offers valuable perspectives for readers in this area. While a significant portion of the cited references dates back approximately five years, the inclusion of up-to-date studies ensures that the citations are appropriate.

The following is minor comments for this MS.

Given that Salmonella is widely recognized as a major cause of foodborne illness, it would be appropriate to note that the strains employed in cancer immunotherapy and clinical trials have been genetically attenuated and therefore present an extremely low risk of inducing conventional food poisoning, as supported by scientific data.

Author Response

We appreciate your valuable feedback, and completely agree with your observation. For this reason, we have emphasized that strains used in cancer immunotherapy are genetically attenuated to minimize pathogenicity, while retaining antitumor properties. The new paragraph has been highlighted in the revised manuscript so that it can be easily located in context (lines 51-56), and it states as follows “It is important to state that an effective bacterial candidate for anticancer therapy must strike an adequate balance between attenuation and antitumor activity. While wild-type strains are not suitable for clinical use due to their ability to induce severe systemic infections, overly attenuated derivatives may show reduced therapeutic efficacy because they fail to elicit a sufficient immune response for tumor regression [5].”.

Reviewer 3 Report

Comments and Suggestions for Authors

The review discusses the role of Salmonella in initiating various cell death modalities. While it is an interesting and valuable contribution, several major issues should be addressed.

1. The review focuses predominantly on the role of Salmonella in myeloid cells. The authors should also include a discussion of Salmonella-induced apoptosis, inflammasome activation, necroptosis, autophagy, and other pathways in lymphocytes.

2. The authors do not specify which types of inflammasomes are affected by Salmonella

3. The review should also address whether Salmonella has functions beyond cell-death–dependent mechanisms in the context of immunotherapy.

4. It would be beneficial to mention whether any clinical trials have investigated Salmonella-based strategies in the context of immunotherapy.

Minor issue:

some sentences lack the references. 

Author Response

We thank the reviewer for the thorough assessment of our work. A detailed point-by-point response to each comment is presented below, and revisions have been clearly indicated in the updated manuscript.

  1. We thank the reviewer for the suggestion to include a broader discussion on Salmonella-induced cell death pathways in lymphocytes. We agree that such mechanisms may contribute in specific contexts. However, our review focuses on myeloid cells because the myeloid compartment constitutes a major proportion of the immune infiltrate in solid tumors, and are the main cellular context in which Salmonella-induced programmed cell death has been experimentally characterized to date. Nonetheless, we accept that inclusion of lymphocyte-related death pathways could improve the discussion and hence, have added a brief paragraph in the revised manuscript to acknowledge this possibility, and highlighting the lack of direct studies in this area, which we consider an important gap and an opportunity for future research. Lines 587-597 now read “Although the focus of this review is on myeloid populations due to their abundance and functional relevance within the TME, it is important to acknowledge that lymphocytes may also undergo Salmonella-induced cell death through different mechanisms. However, the current evidence specifically addressing these pathways in lymphoid cells during Salmonella-based cancer therapies remains limited. For instance, a report suggests that Salmonella Typhi can induce apoptosis in lymphoma-derived T cells in vitro and contributes to tumor regression in murine T-cell lymphoma models [122]. However, in this case, lymphocytes are not other than tumor cells. Another set of studies show that Salmonella inhibits both pyroptosis [123], [124] and autophagy [125] in B cells to persist, but those reports do not address bacteria as cancer therapy. This knowledge gap represents an opportunity for future investigation to clarify whether and how distinct lymphocyte subsets undergo programmed cell death following bacterial targeting in the TME.”
  2. To the best of our knowledge, all particular inflammasomes that have been reported to be activated by Salmonella are specified in the manuscript, in Pyroptosis section (lines 228-239).
  3. We appreciate the reviewer’s comment. We acknowledge that Salmonella also exerts antitumor functions beyond cell-death–dependent mechanisms. However, those aspects have been extensively reviewed elsewhere and a detailed analysis would exceed the scope of the present review, which specifically focuses on Salmonella-induced cell death pathways. A clarifying sentence has now been added to the Introduction (lines 83–85) which reads “Although Salmonella exerts diverse antitumor activities beyond direct induction of tumor cell death, these aspects have been extensively covered elsewhere [15], [16] and fall outside the specific scope of this review.”.
  4. We agree that clinical trial information regarding the use of Salmonella in cancer immunotherapy is valuable. Initially, we chose not to include it because the reported clinical outcomes were not directly linked to cell death mechanisms, which is the main focus of this review. Nonetheless, following the reviewer’s suggestion, we have now added a paragraph addressing available clinical trials in the Introduction section (lines 69–82), which reads “Beyond preclinical evidence, some attenuated Salmonella strains have already been evaluated in clinical settings, supporting the feasibility of exploiting the bacterium as an antitumoral agent. The first-in-human Phase I clinical trial administering intravenous attenuated S. Typhimurium VNP20009 to patients with metastatic melanoma demonstrated that treatment was safe at low doses, with occasional tumor colonization observed and evidence of immune activation, although without objective clinical responses [12]. A subsequent study using continuous intravenous infusion of the same strain further confirmed tolerability, setting safety parameters for future interventions [13]. More recently, an oral formulation engineered to express human IL-2 was tested in patients with metastatic gastrointestinal cancer, also showing an acceptable safety profile and preliminary immune engagement, thus reopening interest in clinical development [14]. Together, these trials highlight that Salmonella-based therapies are clinically implementable and provide a translational framework for next-generation tumor-targeting strains optimized for enhanced safety, delivery, and antitumor efficacy.”

Round 2

Reviewer 3 Report

Comments and Suggestions for Authors

The authors  state that their study focuses on the myeloid compartment, yet the title does not reflect this. The authors should revise the title to include this relevant information.

Author Response

We understand the reviewer´s concern but feel that the title is appropriate and not misleading. To solve this issue and ensure that the scope is immediately clear to readers, we have now revised the Abstract to explicitly state that the review predominantly addresses Salmonella-induced cell death mechanisms in the myeloid compartment, due to the scarcity of mechanistics data in lymphoid lineages.

New lines are highlighted in the updated manuscript that you can find attached here,  and now lines 19-21 state "Given that most mechanistics evidence on Salmonella-induced cell death has been generated in myeloid cells, we primarily focus on the myeloid compartment, while integrating available observations from tumor cells and other immune populations when relevant...".
